# Physical activity during pregnancy and offspring cardiovascular risk factors: findings from a prospective cohort study

Louise A C Millard,[1,2] Debbie A Lawlor,[1,2] Abigail Fraser,[1,2] Laura D Howe[1,2]

[1]MRC Integrative Epidemiology Unit at the University of Bristol, Bristol, UK
[2]School of Social and Community Medicine, University of Bristol, Bristol, UK

**Correspondence to**
Louise A C Millard; louise.millard@bristol.ac.uk

## ABSTRACT

**Objectives:** The long-term consequences of maternal physical activity during pregnancy for offspring cardiovascular health are unknown. We examined the association of maternal self-reported physical activity in pregnancy (18 weeks gestation) with offspring cardiovascular risk factors at age 15.

**Design:** Prospective cohort study.

**Setting:** The Avon Longitudinal Study of Parents and Children (ALSPAC).

**Participants:** 4665 maternal-offspring pairs (based on a sample with multiple imputation to deal with missing data) from the ALSPAC, a prospective cohort based in the South West of England with mothers recruited in pregnancy in 1991–1992.

**Primary and secondary outcome measures:** Offspring cardiovascular risk factors at age 15; body mass index (BMI), waist circumference, systolic blood pressure, diastolic blood pressure, glucose, insulin, low-density lipoprotein cholesterol, high-density lipoprotein cholesterol and triglycerides.

**Results:** Greater maternal physical activity was associated with lower BMI, waist circumference, glucose and insulin in unadjusted analyses. The magnitude of associations was generally small with wide CIs, and most associations attenuated towards the null after adjusting for confounders. The strongest evidence of association after adjustment for confounders was for glucose, although the 95% CI for this association includes the null; a one SD greater physical activity during pregnancy was associated with a −0.013 mmol/L difference in offspring glucose levels (equivalent to approximately one-third of a SD; 95% CI −0.027 to 0.001 mmol/L).

**Conclusions:** Our results suggest that maternal physical activity in pregnancy, measured at 18 weeks gestation, is unlikely to be an important determinant of later offspring cardiovascular health. There was some suggestion of association with offspring glucose, but given that all other associations (including insulin) were null after adjustment for confounders, this result should be interpreted with caution.

## ARTICLE SUMMARY

### Strengths and limitations of this study

- The large sample size, prospective nature of data collection and the detailed measurement of a wide range of offspring cardiovascular risk factors in adolescence.
- The data on maternal physical activity during pregnancy were self-reported, and thus measurement error is possible.
- We cannot assess women's physical activity levels before pregnancy, or the extent to which women changed their physical activity levels between the 18 week questionnaire and the end of the pregnancy. Thus it is possible that physical activity later in pregnancy may be a more important exposure for offspring cardiovascular risk.

on both mother and offspring.[1] Current policy encourages physical activity during pregnancy, with UK guidelines recommending at least 30 min of moderate intensity activity at least four times a week,[2–4] and USA guidelines recommending 2.5 h each week.[5] However, the evidence regarding potential risks and benefits of physical activity during pregnancy is unclear, with a Cochrane systematic review of randomised controlled trials concluding that regular aerobic exercise during pregnancy (mostly swimming, static cycling and floor exercises) improves women's physical fitness, but that there was insufficient evidence to draw firm conclusions about any other short-term or long-term risks or benefits to the woman or her offspring.[1]

Physical activity is associated with lower cardiovascular risk in both adults and children.[6–8] It is possible that greater levels of physical activity during pregnancy might improve later offspring cardiovascular health, either through direct intrauterine mechanisms influencing musculoskeletal or other system development or epigenetic changes, or because mothers who are more active in

## INTRODUCTION

There is increasing interest in the potential effects of physical activity during pregnancy

pregnancy are also likely to be so postnatally and might encourage their children to be more active.[9–12] There is some evidence that greater levels of physical activity during pregnancy is associated with later offspring growth and lower adiposity.[13–16] All but one of these studies[16] had a very small sample size, and none were able to assess cardiovascular risk factors other than adiposity in offspring. Furthermore, a recent study[17] found evidence of an adverse effect of physical activity measured at 30 weeks gestation on high-density lipoprotein cholesterol (HDLc), diastolic blood pressure and body mass index (BMI). Thus, it is unclear whether variation in levels of physical activity during pregnancy is (either beneficially or detrimentally) related to long-term cardiovascular health in offspring.

The aim of this study is to investigate the association of physical activity during pregnancy with a range of offspring cardiovascular risk factors (BMI, waist circumference, glucose, insulin, lipids and blood pressure). We further aim to explore whether any observed associations are explained by measured confounders, or if not, whether they are mediated by intrauterine characteristics such as hypertensive disorders of pregnancy (HDP), gestational diabetes or gestational weight gain or whether they are mediated by the offspring's own later physical activity levels.

## METHODS

The Avon Longitudinal Study of Parents and Children (ALSPAC) is a prospective, population-based study investigating environmental and other factors that affect the health and development of children. The study methods are described in detail elsewhere[18 19] and on the study website (http://www.alspac.bris.ac.uk). In brief, all pregnant women living in the three health districts centred around Bristol, in the South West of England, who had an expected delivery date between 1 April 1991 and 31 December 1992 were invited to take part in the study.

From the 14 062 live-births in the study, we restricted analyses to singleton pregnancies of offspring alive at 1 year, and excluded 1234 women who did not provide data on physical activity during pregnancy, 1 woman with an implausible value of physical activity during pregnancy and 7063 offspring without data for at least one outcome, yielding a sample size of 4665 mother-offspring pairs (see figure 1 for the study participant flow diagram).

### Physical activity during pregnancy

At 18 weeks gestation women were asked, in a validated questionnaire,[20] to report the h/week that they currently spend doing each of 11 types of leisure time physical activities (brisk walking, swimming, antenatal exercise, keep fit exercise, cycling, aerobics, tennis, yoga, jogging, weight training, squash and other exercise). Based on the responses to these questions, we estimated a weighted activity index (ie, the weekly total energy expenditure

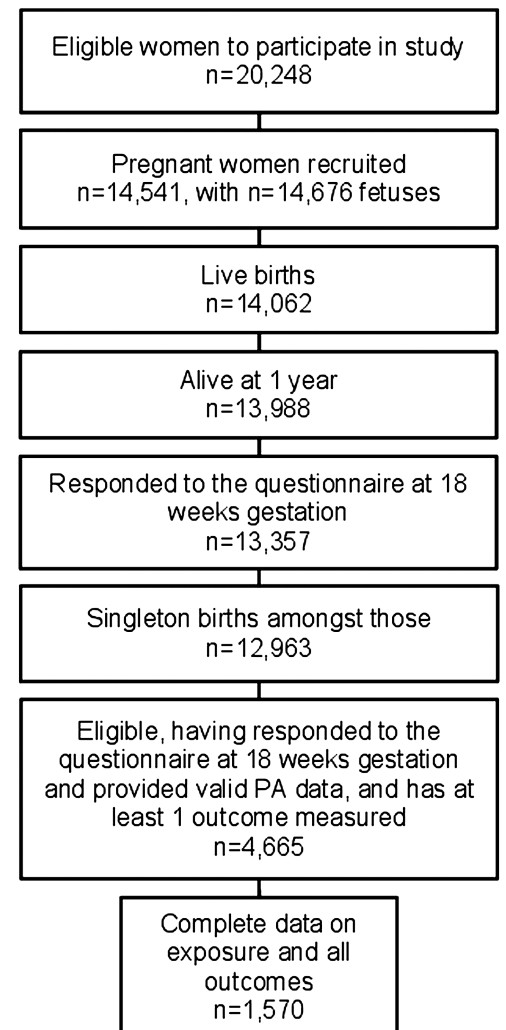

**Figure 1** Flow chart of Avon Longitudinal Study of Parents and Children study participants.

from these activities) by multiplying the published average metabolic equivalent (MET) scores for the reported activities[21] with the estimated h/week (7 for '≥7 h', 4 for '2–6 h', 0.5 for '<1 h' and 0 for 'never')—see online supplementary table S1 for example, MET scores. MET scores are defined as the ratio of the work metabolic rate to a standard resting metabolic rate of 1.0 kcal/kg/h.[22] MET scores were derived from published sources that measured the oxygen cost of each activity, or estimated from the MET scores of similar activities where no studies could be found for this activity.[22] Published MET values are not available for 'antenatal exercise' and 'keep-fit exercise', thus we assigned 2.0 and 3.5 METs, respectively. Physical activity equating to a score greater than 70 was reported by 46 individuals, which is equivalent to 7 h/week of jogging; these values were recoded to 70 in order to prevent them from having undue influence on the results.[22]

### Cardiovascular risk factors in the offspring

Offspring cardiovascular risk factors (BMI, waist circumference, glucose, insulin, lipids (HDLc and low-density

lipoprotein cholesterol (LDLc)) and blood pressure) were measured at a research clinic attended at mean age 15.5 years. Weight and height were measured with the child in light clothing without shoes. BMI was calculated as weight in kilograms divided by height in meters squared. A flexible tape was used to measure waist circumference to the nearest 1 mm at the mid-point between the lower ribs and the pelvic bone. Blood pressure was measured with a Dinamap 9301 Vital Signs Monitor (Morton Medical, London) with the correct cuff size. Two readings of systolic and diastolic blood pressure were recorded, with the child at rest and their arm supported and we used the mean of the two measures. A fasting blood sample was taken and used to measure total cholesterol, triglycerides and HDLc. LDLc concentrations were determined from these with the Friedwald equation (LDLc=total cholesterol−HDLc+triglycerides×0.45 mmol/L)). Insulin was measured with an ELISA (Mercodia, Uppsala, Sweden) that does not cross react with proinsulin and plasma glucose was measured with an automated assay. Further details are provided in Online Resource.

### Other measurements

We considered the following to be potential confounding factors: household social class, maternal education, ethnicity, parity (the number of previous pregnancies resulting in a live birth or stillbirth), smoking during pregnancy, history of hypertension outside of pregnancy, prepregnancy BMI and gestational weight gain in the first 18 weeks of pregnancy. We also adjusted for the offspring's gender and age at assessment of outcomes in all analyses.

We considered the following to be potential mediating factors: HDP (mothers with pre-eclampsia or gestational hypertension), gestational weight gain (between 18–28 weeks and 28 weeks onwards; see Online Resource for explanation of why these two time points were used), gestational diabetes, offspring gestational age, birthweight and offspring accelerometer-assessed moderate to vigorous physical activity at 14 years. Gestational age and birthweight are plausible mediators of the intrauterine pathways due to their association with both gestational diabetes and HDP, and offspring outcomes.[10 11] Details of the assessment of all of these potential confounders and mediators are provided in Online Resource.

### Statistical analysis

All analysis was performed with Stata V.11.2.[23] Insulin and triglycerides were positively skewed and so were log transformed. Multivariable linear regression was used to examine the association between the weighted score of maternal physical activity during pregnancy and offspring cardiovascular risk factors. The weighted score of maternal physical activity in pregnancy was standardised to have a mean of zero and variance of one. Coefficients from regression models therefore represent the mean difference in cardiovascular risk factors (in their

measured units or a percentage difference for the log transformed insulin and triglyceride outcomes) for a one SD increase in MET physical activity, which is equivalent to approximately 15 MET h/week, that is, just over 2 h of jogging or just under 4 h of brisk walking. We also repeated all analyses with each cardiovascular outcome scaled per SD of their distribution (or logged distribution for insulin and triglycerides) and present these results in figures. This allowed us to compare the magnitudes of any associations across outcomes. We used tests of interaction to assess whether the associations between maternal physical activity during pregnancy and offspring cardiovascular risk factors differed by child gender or by maternal prepregnancy BMI (categorised as underweight/normal weight (BMI less than 25) or overweight/obese (BMI 25 or above)).[24]

Figure 2 illustrates the hypothesised pathways between maternal physical activity during pregnancy and offspring cardiovascular risk factors. We carried out several analyses to explore these pathways. We initially investigated minimally adjusted associations, adjusting only for the child's gender and exact age at outcome measurement. Confounder-adjusted models adjusted for child gender, exact age at outcome measurement, household social class, maternal education, ethnicity, parity, smoking during pregnancy, previous hypertension, prepregnancy weight and gestational weight gain between 0 and 18 weeks gestation. The possible mediating mechanisms involved were investigated in two further models: (1) all confounders plus intrauterine characteristics—HDP, gestational diabetes and gestational weight gain after 18 weeks, gestational age and birthweight and (2) all confounders plus offspring physical activity (measured objectively using an accelerometer) at age 14. A final model included all potential confounding and mediating factors.

### Missing data

We used multiple imputation with chained equations (details in Online Resource) to impute missing data for all mother-offspring pairs where the data on maternal physical activity during pregnancy (the exposure) and at least one cardiovascular risk factor at age 15 was observed (the eligible sample, N=4665). Analyses conducted using this imputation dataset were considered our main analyses. However, in order to assess the impact of missing data on our results, we compared the results of the main analysis (using imputed datasets) with results from a complete case approach in which we included only individuals with complete data for all outcomes, all confounders and all mediators. Characteristics of participants and levels of missing data for each variable are shown in online supplementary table S2.

### RESULTS

There was large interindividual variation in reported MET h/week at 18 weeks gestation. Approximately one-third of mothers reported less than 6 MET h/week,

**Figure 2** Pathway illustration for the potential associations between physical activity during pregnancy and offspring cardiovascular risk factors.

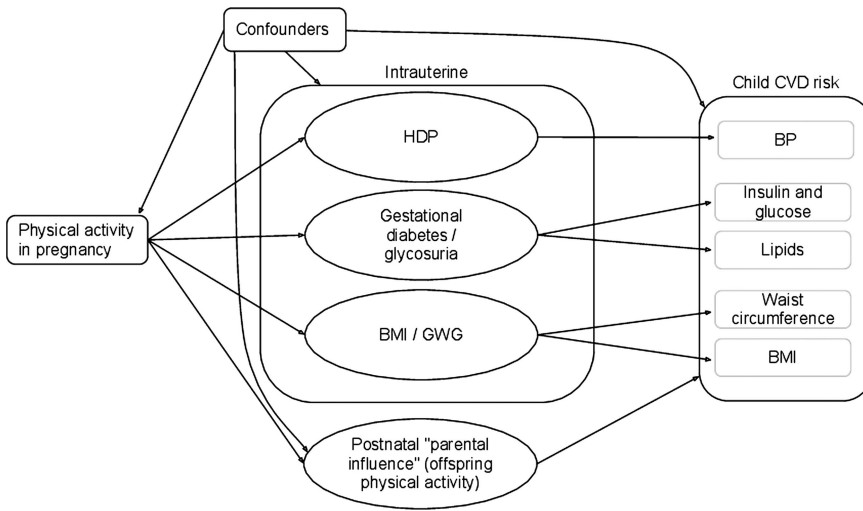

which equates to 1 h of swimming, while approximately one-quarter of women reported over 24 MET h/week. On average, higher social class, greater maternal education, lower parity, no maternal smoking in pregnancy and lower maternal prepregnancy BMI were associated with greater reported physical activity at 18 weeks gestation (table 1). There was limited evidence that maternal physical activity was associated with most of our a priori specified mediators of the association between physical activity in pregnancy and offspring cardiovascular risk factors; no strong evidence of association was seen for HDP, diabetes, gestational weight gain, gestational age or birth weight, but greater maternal physical activity during pregnancy was associated with greater offspring physical activity at age 14. For each SD increase in reported physical activity during pregnancy, there was a 0.06 SD increase in physical activity in the offspring (95% CI 0.03 to 0.09; see online supplementary table S3).

### Association between physical activity during pregnancy and offspring cardiovascular risk factors

There was no evidence that the associations between maternal physical activity during pregnancy and offspring cardiovascular risk factors differed either by offspring gender (all p for interaction >0.2) or by maternal prepregnancy BMI status (all p for interaction >0.1 see online supplementary figure S1). Figure 3 and table 2 show the minimally adjusted, confounder adjusted and mediator (intrauterine characteristics and offspring physical activity) adjusted associations of maternal physical activity during pregnancy with offspring cardiovascular risk factors. More detailed results from models examining each potential mediator separately are given in online supplementary table S4.

There was little evidence that maternal physical activity during pregnancy was associated with the offspring cardiovascular risk factors. In minimally adjusted analyses (adjusting only for child gender and age at outcome assessment), higher levels of physical activity were associated with lower offspring BMI, smaller waist circumference, lower glucose and lower insulin. For all outcomes except insulin, glucose, BMI and waist circumference, the magnitude of coefficients was small and CIs were wide. A one SD increase in physical activity during pregnancy is associated with a lower insulin level in the offspring of −1.90% (95% CI −3.50 to −0.30) and lower glucose level by −0.013 mmol/L (95% CI −0.027 to 0.001). Adjustment for measured confounders moved most of the associations towards the null. After adjusting for confounding, one SD increase in physical activity during pregnancy is associated with lower offspring insulin by −1.20% (95% CI −2.80 to 0.40). The borderline association with glucose did not alter after adjusting for confounders. Adjusting for potential mediators had little effect on the estimated associations for any outcome.

### Sensitivity of results to missing data
The results based on multiple imputation were very similar to those based on individuals with complete data (see online supplementary figure S2 for BMI, online supplementary figure S3 for insulin and online supplementary figure S4 for glucose are shown as illustrative examples, in Online Resource).

### DISCUSSION
Despite greater maternal physical activity in pregnancy being associated with lower prepregnancy BMI and greater offspring physical activity at mean age 14 years, there was no strong evidence that it was related to later offspring cardiovascular risk factors (BMI, waist circumference, blood pressure (systolic blood pressure and diastolic blood pressure), lipids (triglycerides, LDLc and HDLc), and fasting glucose or insulin) at age 15 years using data from a contemporary general population pregnancy cohort. The associations we observed were small, and were generally attenuated with adjustment for

**Table 1** Association of participant characteristics with maternal physical activity during pregnancy: using multiply imputed data (N=4665) of individuals with data for maternal physical activity (exposure) and at least one outcome

| Potential confounder | Confounder category | SD change of maternal physical activity during pregnancy associated with a 1 SD increase of confounder, with 95% CI |
|---|---|---|
| Household social class* | Class I (professional) | Reference |
| | Class II | 0.00 (−0.08 to 0.09) |
| | Class III non-manual | −0.18 (−0.27 to −0.09) |
| | Class III manual | −0.16 (−0.28 to −0.04) |
| | Class IV/V (manual) | −0.15 (−0.31 to 0.01) |
| Maternal education† | Less than O-level | Reference |
| | O-level | 0.17 (0.09 to 0.25) |
| | A-level | 0.39 (0.30 to 0.47) |
| | Degree or above | 0.40 (0.30 to 0.49) |
| Ethnicity‡ | Non-white | Reference |
| | White | −0.11 (−0.25 to 0.03) |
| Parity§ | 0 | Reference |
| | 1 | −0.05 (−0.11 to 0.02) |
| | 2+ | −0.10 (−0.19 to −0.02) |
| Maternal smoking in pregnancy | No | Reference |
| | Yes | −0.09 (−0.17 to −0.01) |
| Previous hypertension | No | Reference |
| | Yes | −0.03 (−0.17 to 0.12) |
| Maternal prepregnancy BMI (SD) | | −0.39 (−0.58 to −0.20) |
| Gestational weight gain 0–18 weeks (SD) | | −0.00 (−0.03 to 0.03) |

*Household social class: The mother recorded the occupation of both herself and her partner in a questionnaire at 32 weeks gestation, which were used to allocate them to social class groups using the 1991 Office of Population, Censuses and Surveys classification; the highest class of the mother and her partner was used in analysis.
†Maternal education: O-level (ordinary level) exams are taken in different subjects usually at age 15–16 at the completion of legally required school attendance, equivalent to today's General Certificate of Secondary Education; A-level (Advanced-level) exams are taken in different subjects usually at age 18.
‡Ethnicity: Mixed was recoded as non-white due to the small numbers in the mixed category (n=3).
§Parity: Obtained from obstetric records. Mothers with parity of two or more were grouped into a single category.
BMI, body mass index.

measured confounders. The strongest evidence of association was for glucose (one SD greater physical activity in pregnancy was associated with a lower glucose by −0.013 mmol/L (95% CI −0.027 to 0.001), equivalent to approximately one-third of a SD), but given that we investigated several outcomes, that the association with all other outcomes including insulin attenuated towards the null after adjustment for confounding, and that no associations were seen for BMI, waist circumference or any other cardiovascular risk factor, this finding should be interpreted with caution and requires replication in other studies.

To our knowledge only one study to date has investigated the association of maternal physical activity during pregnancy with a range of offspring cardiovascular risk factors.[17] This study included 965 Danish participants at age 20, and examined the associations of self-reported physical activity at work, leisure time physical activity, daily amount of walking or cycling and sport participation between 18 and 30 weeks gestation (measured at 30 weeks gestation), with offspring cardiovascular risk factors. The authors found that higher levels of self-reported walking-cycling at 30 weeks gestation was associated with higher offspring BMI and diastolic blood pressure, lower HDLc and (in males only) higher systolic blood pressure and waist circumference. The associations were weak and of small magnitude. No associations were seen for the other aspects of physical activity assessed in the analysis. It is therefore possible that the associations observed in this study occurred due to chance. However, the authors' conclusions are in line with our own, in that there was no evidence to support the a priori hypothesised potential benefit of physical activity during pregnancy for offspring cardiovascular health.

The main strengths of our study are the large sample size, prospective nature of data collection, and the detailed measurement of a wide range of offspring cardiovascular risk factors in adolescence. The measure of maternal physical activity during pregnancy uses a

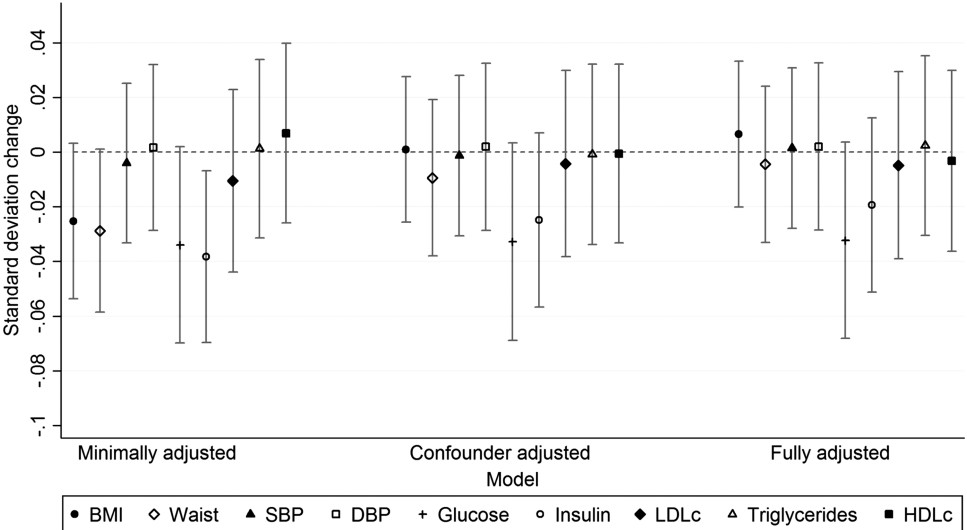

**Figure 3** Associations of maternal physical activity during pregnancy with offspring cardiovascular risk factors: using multiply imputed data (N=4665) of individuals with data for maternal physical activity (exposure) and at least one outcome. Mean SD difference (and 95% CIs) in offspring cardiovascular risk factors (or logged distribution for insulin and triglycerides) for a one SD increase in metabolic equivalent maternal physical activity. Minimally adjusted: adjusted for child gender and child age at 15 year clinic. Confounder adjusted: adjusted for child age at 15 year clinic, child gender, socioeconomic position confounders (household social class, maternal education, ethnicity, parity, mother smoker and previous hypertension), maternal prepregnancy BMI (weight and height) and gestational weight gain 0–18 weeks. Fully adjusted with adjustment for all confounders and mediators: confounder adjusted with additional adjustment for intrauterine mechanisms (hypertensive disorders of pregnancy (none, gestational hypertension, pre-eclampsia), diabetes/glycosuria (none, existing diabetes, gestational diabetes, glycosuria), gestational weight gain 18–24 weeks, gestational weight gain 24 weeks onwards), birth weight, gestational age and child's physical activity at age 14.

weighted index, which attempts to measure the energy expenditure of individuals using previously published average values for different types of physical activity (see online supplementary table S1). We were able to assess the extent to which our observed associations are confounded by maternal prepregnancy BMI and socioeconomic factors, and explore a number of potential a priori specified mediating pathways (intrauterine factors and shared familial environment, represented by the child's own physical activity). Additionally, we used multiple imputation to examine and correct for potential bias introduced by missing data.

The results using complete case data and using multiply imputed data were similar. In both analyses, the conclusion remains that there is no strong association between reported levels of physical activity during pregnancy and offspring BMI (or any other cardiovascular risk factor). Multiple imputation gives unbiased results providing the data are missing at random, that is, assuming that the missingness is not related to the value of the observation, conditional on the variables included in the imputation model. We included in the imputation models a wide range of variables predictive of both missingness in the outcomes and of levels of cardiovascular risk factors in those with observed data, including socioeconomic factors, maternal and family characteristics and earlier measures of the offspring's cardiovascular disease risk factors, in order to make sure that this

assumption holds. The loss to follow-up in our study would result in selection bias if the association between maternal physical activity during pregnancy and offspring cardiovascular risk differed between those included in our analyses and those lost to follow-up. However, various studies have investigated the potential effects of loss to follow-up on associations in epidemiological studies, and have found that they are generally robust to even severe loss to follow-up.[25–28]

A limitation of our work is that the data on maternal physical activity during pregnancy were self-reported, and thus measurement error is inevitable. It is possible that the women may under-report or over-report their physical activity levels but we do not have objective data with which to evaluate this for our participants.[20] However, associations of socioeconomic position, prepregnancy BMI, maternal education and objectively measured offspring physical activity at age 14 were as expected, providing some evidence of face validity. MET scores may not accurately reflect the energy expenditure of physical activity in pregnancy, as they are not derived from studies of pregnant individuals. However, this is inconsequential as we use this measure as an activity classification scheme rather than an exact quantification of energy expenditure,[29] and MET scores have been used in this way in several previous studies.[30 31] The 'other' activity category was selected by 9.45% of the women in our study and it is possible this may reduce

**Table 2** Association between maternal physical activity during pregnancy and offspring cardiovascular risk factors: using multiply imputed data (N=4665) of individuals with data for maternal physical activity (exposure) and at least one outcome

| Outcome | Change per 1 SD greater maternal metabolic equivalent physical activity (95% CIs) | | |
| --- | --- | --- | --- |
| | Minimally adjusted | Confounder adjusted | All confounders and mediators |
| Mean difference (in units specified in first column) per 1 SD greater maternal physical activity during pregnancy | | | |
| BMI, kg/m² | −0.089 (−0.190 to 0.011) | 0.004 (−0.091 to 0.098) | 0.023 (−0.071 to 0.117) |
| Waist circumference, cm | −0.271 (−0.552 to 0.011) | −0.088 (−0.358 to 0.181) | −0.042 (−0.311 to 0.227) |
| SBP, mm Hg | −0.044 (−0.360 to 0.272) | −0.014 (−0.332 to 0.303) | 0.017 (−0.301 to 0.334) |
| DBP, mm Hg | 0.015 (−0.251 to 0.280) | 0.017 (−0.251 to 0.284) | 0.018 (−0.250 to 0.286) |
| Glucose, mmol/L | −0.013 (−0.027 to 0.001) | −0.013 (−0.027 to 0.001) | −0.012 (−0.026 to 0.001) |
| LDLc, mmol/L | −0.006 (−0.025 to 0.013) | −0.002 (−0.022 to 0.017) | −0.003 (−0.022 to 0.017) |
| HDLc, mmol/L | 0.002 (−0.008 to 0.012) | 0.000 (−0.010 to 0.010) | −0.001 (−0.011 to 0.009) |
| Percentage difference per 1 SD greater maternal physical activity during pregnancy | | | |
| Insulin, percentage of difference | −1.90 (−3.50 to −0.30) | −1.20 (−2.80 to 0.40) | −1.00 (−2.60 to 0.60) |
| Triglycerides, percentage of difference | 0.00 (−1.20 to 1.30) | 0.00 (−1.30 to 1.20) | 0.00 (−1.20 to 1.40) |

Results of additional models with adjustment for mediators are in online supplementary table S4.
Minimally adjusted: Adjusted for child gender and child age at 15 year clinic.
Confounder adjusted: Adjusted for child age at 15 year clinic, child gender, maternal and family confounders (household social class, maternal education, ethnicity, parity, mother smoker, previous hypertension, maternal prepregnancy BMI (weight and height) and gestational weight gain 0–18 weeks).
All confounders and mediators: Confounder adjusted with additional adjustment for intrauterine mechanisms (hypertensive disorders of pregnancy (none, gestational hypertension, pre-eclampsia), diabetes/glycosuria (none, existing diabetes, gestational diabetes and glycosuria), gestational weight gain 18–24 weeks, gestational weight gain 24 weeks onwards), birth weight, gestational age and child's physical activity at age 14.
BMI, body mass index; DBP, diastolic blood pressure; HDLc, high-density lipoprotein cholesterol; LDLc, low-density lipoprotein cholesterol; SBP, systolic blood presure.

the strength of the associations, as several activity types with different energy expenditures are assigned the same MET score. We cannot assess women's physical activity levels before pregnancy, or the extent to which women changed their physical activity levels during pregnancy, both before and after the 18 week question-naire, as this measure of physical activity was only cap-tured once in this cohort. Previous research with this cohort has shown that two-thirds of women report at 18 weeks gestation that they have reduced their physical activity levels since becoming pregnant, but at 18 and 32 weeks gestation similar proportions of women report at least 3 h/week of strenuous physical activity (defined as 'sufficient to cause sweating').[22] Thus it is possible that physical activity either earlier or later in pregnancy may be a more important exposure for offspring cardio-vascular risk—further studies are necessary to test this hypothesis.

Although we demonstrated no evidence of association between physical activity during pregnancy and off-spring cardiovascular risk factors, we did find that greater levels of physical activity during pregnancy were associated with greater levels of objectively assessed physical activity in the offspring at mean age 14 years. This measure of offspring physical activity has been shown to be associated with fat mass and cardiovascular risk factors.[8 32]

Our results, in line with the only existing similar study to date,[17] suggest that maternal physical activity in preg-nancy is unlikely to be an important determinant of later offspring cardiovascular health, though further large prospective studies of this association are needed to be confident that this is the case, particularly with studies using objectively measured physical activity at various times throughout pregnancy.

**Correction notice** This article has been corrected since it was published Online First. The Open Access statement has been corrected.

**Contributors** LACM contributed to the design of the study, performed the analyses, drafted the initial manuscript, reviewed and revised the manuscript and approved the final version of the manuscript as submitted. DAL conceptualised and designed the study, critically reviewed and revised the manuscript and approved the final version of the manuscript as submitted. AF contributed to the design of the study, critically reviewed and revised the manuscript and approved the final version of the manuscript as submitted. LDH contributed to the design of the study, supervised the analyses, critically reviewed and revised the manuscript and approved the final version of the manuscript as submitted.

**Funding** The UK Medical Research Council (grant G074882), the Wellcome Trust (grant WT076467) and the University of Bristol provide core funding support for ALSPAC. The UK Medical Research Council (grant G0600705) and the University of Bristol provide core funding for the Medical Research Council Centre for Causal Analyses in Translational Epidemiology.

**Competing interests** LACM is funded by a studentship from the UK Medical Research Council. LDH was funded by a UK Medical Research Council Population Health Scientist Fellowship (G1002375). AF is funded by a UK Medical Research Council research fellowship (grant 0701594).

**Ethics approval** ALSPAC Law and Ethics Committee and the Local Research Ethics Committees.

**Provenance and peer review** Not commissioned; externally peer reviewed.

**Data sharing statement** The ALSPAC policy on data sharing is available on the website: http://www.bristol.ac.uk/alspac. To discuss access to ALSPAC data, please contact the ALSPAC executive team on alspac-exec@bristol.ac.uk.

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
