## [Reviewer comments · BMJ Open]

Some articles will have been accepted based in part or entirely on reviews undertaken for other BMJ Group journals. These will be reproduced where possible.

ARTICLE DETAILS

TITLE (PROVISIONAL)	Physical activity during pregnancy and offspring cardiovascular risk factors: findings from a prospective cohort study
AUTHORS	Millard, Louise; Lawlor, Debbie; Fraser, Abigail; Howe, Laura

VERSION 1 - REVIEW

REVIEWER	Inge Danielsen MSc, MPA, PhD student Research coordinator Rigshospitalet Diabetes and Metabolism, Denmark I have no competing interests.
REVIEW RETURNED	01-Aug-2013

THE STUDY	Much detailed information concerning methods is only available in the supplementary material. I would highly appreciate it in the paper. However, it may be the policy of the journal to accept it this way. Moreover, the exposure measure (the level of maternal physical activity) should be discussed further as i mention in my comments for the authors.
GENERAL COMMENTS	This prospective study examined the relationship between maternal physical activity assessed at 18 weeks of gestation and offspring cardiovascular risk factors at 15 years of age. Results overall found no significant relationship. This is one of the first studies to explore correlations between maternal prenatal physical activity and offspring metabolic risk. The cohort is large and allows high statistical power. The analyses are extensive and thorough, and the paper is well written with a clear thread of consistency. The method section is limited in important details, which however can be found in the supplementary material. 1. Page 2, line 50: Please specify, which confounders were included in the analyses? 2. Page 2, line 55: The confidence interval is narrow, however includes 0. If the association is mentioned, it should be underlined that the association is only borderline.

3. Page 3, line 8: It doesn't make sense to mention insulin here for the first time.

4. Page 5, second paragraph, second sentence: Have you considered epigenetic mechanisms?

5. Page 5, second paragraph: I suggest you rephrase and get rid of the brackets.

6. Page 6: Has the questionnaire been validated?

7. Page 6: Include details regarding MET scores and MET hours. How were the MET scores originally generated – from with data? Is this measure applicable for pregnant women?

8. Page 6, line 54: Which lipids?

9. Page 6, bottom line: "Further details are provided in Online Resource": Some details are desirable in the text.

10. Page 8, line 19: How was offspring PA measured? Should to some extent be explained in the text in addition to a reference to the online resources.

11. Page 9: Have you analyzed the magnitude of selection bias – what does the relatively high loss to follow up mean?

12. Page 10: The association with glucose remains borderline significant.

13. Page 11: Regarding the size of the association: approximately one third of a standard deviation – could this have physiological consequences for the offspring? Is it possible and can we know?

	14. Page 11: The cited study (ref. 17) measures maternal PA during approximately weeks 18-30 by questionnaires answered by the women in gestation week 30 (the women were asked about their PA during the previous three months). 15. Page 12, line 32: measurement error is likely – it's inevitable! Please discuss the quality of the maternal PA measure. Are the women likely to over-report or under-report? Can we use the MET-scores for pregnant women? Will the physical activity level of an average women change during pregnancy and how could that affect the potential associations with offspring metabolic markers? How many women (in percentage of the total women included in the analyses) answered "Other" when asked about their physical activity? And how was the MET score for "Other" assessed? Does this affect the (lack of) associations? The maternal PA is the central subject of the paper – it should be discussed more. 16. Page 12, line 34: Please give information on the objectively measured offspring PA in the Methods section. 17. Supplementary Table 1: This table is not completely self-explained. Please provide more info. You could show an additional column with the MET hours (the two existing columns multiplied). In the footnote you must mean MET hours and not MET score?
--	---

REVIEWER	Alicia Matijasevich Professora Adjunta do Departamento de Medicina Social e do Programa de Pós-Graduação em Epidemiologia Universidade Federal de Pelotas (UFPel) Brasil No competing interests.
REVIEW RETURNED	08-Aug-2013

THE STUDY	Supplemental material presented is fine as it is.
GENERAL COMMENTS	The manuscript is well written and fulfills all the conditions to be considered a high quality paper. Just minor comments: Page 7, 2nd paragraph: please explain what HDP is the first time you mention it. Page 9, last paragraph: I would include BMI and waist circumference in the phrase "For all outcomes except insulin and glucose, the magnitude of coefficients was small and confidence intervals were

	wide". The difference between BMI, waist circumference, glucose and insulin is very small (look at figure 1, minimally adjusted analysis). Figure 2: I found this figure a bit confusing. HDP, gestational diabetes/glycosuria and BMI/GWG come before "postnatal parental influence (offspring physical activity)" and could affect offspring physical activity too. Table 2: I think it is a typographical error: parity, maternal smoking during pregnancy and previous hypertension were included inside "socioeconomic position confounders"
--	--

REVIEWER	Jeremy Pomeroy, MS, PhD Intramural Research and Training Award Fellow National Institute of Diabetes and Digestive and Kidney Diseases National Institutes of Health USA I have no conflicts of interest.
REVIEW RETURNED	12-Aug-2013

THE STUDY	Statistical Methods Overall the manuscript is very clear and concise, the only exception is the description and discussion of the mediation analysis. The first full paragraph on page 8 (lines 14-21) includes a description of the two regression models that were used to investigate possible mediation. No formal test of mediation (or indirect) effect is described or shown. Please provide more specific information how possible mediation was formally assessed. Supplemental documents All of the supplemental material provided in the supplemental document is appropriate as an appendix and is not necessary to be included in the manuscript.
GENERAL COMMENTS	Minor points 1) The discussion and conclusion clearly reflect the limitation that the only maternal physical activity data is from self-report at 18 weeks gestation. In the conclusion section of the abstract I suggest adding that the conclusion is about maternal physical activity at 18 weeks gestation. 2) Related to point 1, the bullet point under "Key messages" in the article summary (pg 4 lines 16-18) should also be qualified with a limitation that physical activity was assessed by self-report at 18 weeks gestation. 3) The discussion (pg 12 line 36-40) includes a statement that physical activity before pregnancy as well as physical activity during the rest of pregnancy after the time covered in the questionnaire could not be assessed. Physical activity during early pregnancy, up to the time covered in the questionnaire, also could not be assessed. This should be stated.

Reviewer: Inge Danielsen
MSc, MPA, PhD student
Research coordinator
Rigshospitalet
Diabetes and Metabolism, Denmark

I have no competing interests.

Much detailed information concerning methods is only available in the supplementary material. I would highly appreciate it in the paper. However, it may be the policy of the journal to accept it this way.

Moreover, the exposure measure (the level of maternal physical activity) should be discussed further as I mention in my comments for the authors.

This prospective study examined the relationship between maternal physical activity assessed at 18 weeks of gestation and offspring cardiovascular risk factors at 15 years of age. Results overall found no significant relationship. This is one of the first studies to explore correlations between maternal prenatal physical activity and offspring metabolic risk. The cohort is large and allows high statistical power. The analyses are extensive and thorough, and the paper is well written with a clear thread of consistency. The method section is limited in important details, which however can be found in the supplementary material.

Response: We thank Ms Danielsen for her comments, and respond to the concerns regarding the detail of the methods in the individual comments below. We have tried to keep the manuscript succinct to appeal to general readership, and have provided more detailed methodological details in the supplementary materials. We have ensured that sufficient detail is provided in the main paper such that it is understandable without reference to the supplemental material, and we feel that keeping the in-depth details of methods in the supplement is beneficial to the readability of the paper. In response to the reviewer's suggestion, we have moved some text to the main paper, e.g. details of the cardiovascular measurements. However, if the editors feel that additional aspects of the supplementary material should be moved to the main paper, we would be happy to do this.

1. Page 2, line 50: Please specify, which confounders were included in the analyses?

Response: We have not added confounders to this part of the abstract, as this would take the word count above that required by BMJ Open.

2. Page 2, line 55: The confidence interval is narrow, however includes 0. If the association is mentioned, it should be underlined that the association is only borderline.

Response: We have amended this to explicitly say the confidence interval includes the null.

3. Page 3, line 8: It doesn't make sense to mention insulin here for the first time.

Response: We appreciate this comment, but believe it is relevant to mention insulin here as the close relationship between glucose and insulin means that any difference in associations does provide an important indication of validity of results.

4. Page 5, second paragraph, second sentence: Have you considered epigenetic mechanisms?

Response: Thank you, we have added the possibility of epigenetic mechanisms to the sentence describing possible ways that physical activity in pregnancy may affect the cardiovascular risk of the

child.

5. Page 5, second paragraph: I suggest you rephrase and get rid of the brackets.

Response: We have changed this sentence to make it clearer, to: "All but one of these studies [16] had a very small sample size, and none were able to assess cardiovascular risk factors other than adiposity in offspring."

6. Page 6: Has the questionnaire been validated?

Response: The questionnaire used to assess physical activity during pregnancy was developed in a study of 100 pregnant women, in which various questionnaires were compared with two established assessment methods for physical exertion. We have now added a reference to this study in our paper.

7. Page 6: Include details regarding MET scores and MET hours. How were the MET scores originally generated – from with data? Is this measure applicable for pregnant women?

Response: We have added further information including the definition of a MET score and how these scores are derived, as well as the reference mentioned above detailing the design and validation of the questionnaire.

8. Page 6, line 54: Which lipids?

Response: We have added the specific lipids – HDLc and LDLc.

9. Page 6, bottom line: "Further details are provided in Online Resource": Some details are desirable in the text.

Response: We have moved some detail about the cardiovascular outcomes from the supplemental material to the main text.

10. Page 8, line 19: How was offspring PA measured? Should to some extent be explained in the text in addition to a reference to the online resources.

Response: We have added details of the offspring PA measurement, which is measured objectively with an accelerometer.

11. Page 9: Have you analyzed the magnitude of selection bias – what does the relatively high loss to follow up mean?

Response: Loss to follow-up would result in selection bias if the association between maternal physical activity during pregnancy and offspring cardiovascular risk differed between those included in our analyses and those lost to follow-up. There is no reason to believe that this would be the case. Various studies have investigated the potential effects of loss to follow-up on associations in epidemiological studies, and have found that they are generally very robust to even very severe loss to follow-up. We have now expanded our discussion of this issue and provided some references to support this statement.

12. Page 10: The association with glucose remains borderline significant.

Response: We have added the fact that it is a borderline association.

13. Page 11: Regarding the size of the association: approximately one third of a standard deviation – could this have physiological consequences for the offspring? Is it possible and can we know?

Response: Levels of cardiovascular risk factors tend to track from childhood to adulthood, so a reduction of one third of a standard deviation in glucose levels may well be important in terms of adult disease risk. However, we do not wish to over-emphasise this finding in the manuscript. As we discuss on page 11, the lack of observed associations for insulin, BMI and waist circumference mean that we believe this finding should be interpreted with caution.

14. Page 11: The cited study (ref. 17) measures maternal PA during approximately weeks 18-30 by questionnaires answered by the women in gestation week 30 (the women were asked about their PA during the previous three months).

Response: Thank you, we have corrected this information.

15. Page 12, line 32: measurement error is likely – it's inevitable! Please discuss the quality of the maternal PA measure. Are the women likely to over-report or under-report? Can we use the MET-scores for pregnant women? Will the physical activity level of an average women change during pregnancy and how could that affect the potential associations with offspring metabolic markers? How many women (in percentage of the total women included in the analyses) answered "Other" when asked about their physical activity? And how was the MET score for "Other" assessed? Does this affect the (lack of) associations?

The maternal PA is the central subject of the paper – it should be discussed more.

Response: We agree that measurement error is inevitable, and welcome the opportunity to expand our discussion of this issue in the manuscript. However, it is important to emphasise that very few population-based studies exist that have objective measures of physical activity during pregnancy, particularly on a large number of women and with long-term follow-up on the offspring. Thus we do not believe that the self-reported nature of our physical activity data undermines the importance of our findings. Some of the key points that we have now added discussion of in the manuscript are:

- Both under- and over-reporting of activity levels are possible, and we do not have any objective data with which to evaluate this in our participants.
- Several previous studies have used MET scores in pregnant women. MET scores do not aim to provide an exact quantification of energy expenditure through physical activity; rather they seek to provide an activity classification scheme that standardises the measurement of physical activity across studies.
- Few studies have sufficient data to examine whether physical activity levels change over pregnancy. However, previous research on this cohort has shown that two thirds of women report in questionnaires at 18 weeks gestation that they have reduced their physical activity levels since becoming pregnant, but a similar proportion of women in this cohort report at least 3 hours per week of strenuous physical activity (defined as 'sufficient to cause sweating') at 18 and 32 weeks gestation.
- The "other" activity category was selected by 9.45% of the women in our study and it is possible this may reduce the association towards the null as several activity types with different energy expenditures are assigned the same MET score. We could not find other studies to compare this with, and have added this to the discussion as a limitation of the paper.

16. Page 12, line 34: Please give information on the objectively measured offspring PA in the Methods section.

Response: We have added further details about offspring PA to the methods section.

17. Supplementary Table 1: This table is not completely self-explained. Please provide more info. You could show an additional column with the MET hours (the two existing columns multiplied). In the footnote you must mean MET hours and not MET score?

Response: The mean MET score of the mothers in this study is 15.8, where a MET score represents the energy expended in 1 hour of each activity. The number of hours equating to the mean MET score is 15.8/MET score, for each activity. Therefore the additional column suggested would equate to the mean MET hours, hence the same for all activities (15.8). We have added further details to make this table clearer, and thank you for your comments.

Reviewer: Alicia Matijasevich
Professora Adjunta do Departamento de Medicina Social e do Programa de Pós-Graduação em Epidemiologia
Universidade Federal de Pelotas (UFPel)
Brasil

No competing interests.

The manuscript is well written and fulfills all the conditions to be considered a high quality paper.

Response: We thank Professor Matijasevich for her comments.

Just minor comments:

Page 7, 2nd paragraph: please explain what HDP is the first time you mention it.

Response: We have added this information, that this is mothers with pre-eclampsia or gestational hypertension.

Page 9, last paragraph: I would include BMI and waist circumference in the phrase "For all outcomes except insulin and glucose, the magnitude of coefficients was small and confidence intervals were wide". The difference between BMI, waist circumference, glucose and insulin is very small (look at figure 1, minimally adjusted analysis).

Response: We agree and have added BMI and waist circumference to this statement.

Figure 2: I found this figure a bit confusing. HDP, gestational diabetes/glycosuria and BMI/GWG come before "postnatal parental influence (offspring physical activity)" and could affect offspring physical activity too.

Response: Thank you for your feedback about this diagram, which visualises the hypothesised pathways we have analysed. We believe that the main driver of associations of HDP, gestational diabetes, glycosuria, BMI and GWG with the child's physical activity is shared confounding by socioeconomic position and obesity-related behaviours. In our original version of this diagram, we omitted to include an arrow from 'confounders' to 'postnatal influence/offspring physical activity', which may be why the reviewer found this diagram confusing. We have now added this arrow.

Table 2: I think it is a typographical error: parity, maternal smoking during pregnancy and previous hypertension were included inside "socioeconomic position confounders"

Response: Thank you, we have rephrased this to read "maternal and family confounders"

Reviewer: Jeremy Pomeroy, MS, PhD
Intramural Research and Training Award Fellow
National Institute of Diabetes and Digestive and Kidney Diseases
National Institutes of Health
USA

I have no conflicts of interest.

Response: We thank Dr Pomeroy for his comments.

Statistical Methods

Overall the manuscript is very clear and concise, the only exception is the description and discussion of the mediation analysis. The first full paragraph on page 8 (lines 14-21) includes a description of the two regression models that were used to investigate possible mediation. No formal test of mediation (or indirect) effect is described or shown. Please provide more specific information how possible mediation was formally assessed.

Response: Adjusting a regression model for a proposed mediator is a test of mediation; if the regression coefficient attenuates towards the null, this is indicative of mediation. More formal techniques such as path analysis can be used to quantify the proportion of the association mediated by a specific variable, but given that we did not find any strong associations between maternal physical activity in pregnancy and offspring cardiovascular risk, we concluded that this technique was not necessary in this paper. One concern with using our method for mediation analysis is that the adjustment for the mediator can induce confounding of the exposure – outcome association by confounders of the mediator-outcome association. Furthermore, the method we used cannot deal with interactions between the exposure and mediator. Techniques such as marginal structural models can be used to investigate mediation if either of these is a particular concern. However, we have no reason to suspect that either issue is relevant to our example, and as above, since we did not observe strong associations between physical activity in pregnancy and offspring cardiovascular risk factors, the mediation analysis is of less importance, and we did not feel it was appropriate to over-complicate the paper with complex methods for mediation analysis.

Supplemental documents

All of the supplemental material provided in the supplemental document is appropriate as an appendix and is not necessary to be included in the manuscript.

Response: We have used supplementary material according to standard journal practice but we would be happy for the material to be included as an appendix if the editors felt that this was more appropriate.

Minor points

1) The discussion and conclusion clearly reflect the limitation that the only maternal physical activity data is from self-report at 18 weeks gestation. In the conclusion section of the abstract I suggest adding that the conclusion is about maternal physical activity at 18 weeks gestation.

Response: We have added this detail to the abstract conclusion.

2) Related to point 1, the bullet point under "Key messages" in the article summary (pg 4 lines 16-18) should also be qualified with a limitation that physical activity was assessed by self-report at 18 weeks gestation.

Response: We have added this detail to the key message.

3) The discussion (pg 12 line 36-40) includes a statement that physical activity before pregnancy as well as physical activity during the rest of pregnancy after the time covered in the questionnaire could not be assessed. Physical activity during early pregnancy, up to the time covered in the questionnaire, also could not be assessed. This should be stated.

Response: We have changed this sentence to refer to both before and after the 18 week questionnaire.

Response: We agree and thank the reviewers for their valuable comments. Changes to meet Journal requirements have been made.